# Communicating Risk Regarding Food Consumption: The Case of Processed Meat

**DOI:** 10.3390/nu11020400

**Published:** 2019-02-14

**Authors:** Slavica Zec, Clara Minto, Carlo Agostoni, Carolina Fano, Honoria Ocagli, Giulia Lorenzoni, Dario Gregori

**Affiliations:** 1Unit of Biostatistics, Epidemiology and Public Health, Department of Cardiac, Thoracic, Vascular Sciences and Public Health, University of Padova, 35131 Padova, Italy; slavica.zec@unipd.it (S.Z.); clara.minto@unipd.it (C.M.); carolina.fano@studenti.unipd.it (C.F.); honoria.ocagli@unipd.it (H.O.); giulia.lorenzoni@unipd.it (G.L.); 2Pediatric Intermediate Care Unit, Fondazione IRCCS Ca’ Granda-Ospedale Maggiore Policlinico, 20122 Milan, Italy; carlo.agostoni@unimi.it; 3Department of Clinical Sciences and Community Health, University of Milan, 20122 Milan, Italy

**Keywords:** processed meat, colorectal cancer, risk communication, risk perception, unemployment

## Abstract

The present research combines real data and parameters found in recent literature that were used to design realistic scenarios demonstrating the potential effects (benefits and costs) of the World Health Organization (WHO)’s risk communication regarding the consumption of processed meat, which was proven to be associated with an increased risk of colorectal cancer (CRC) in an International Agency for Research on Cancer (IARC)/WHO report. The impact of the risk communication of processed meat consumption was simulated using Monte Carlo microsimulation models. The results showed that a 1% reduction in the number of high-level processed meat consumers may lead to a yearly decrease in CRC cases of 406.43 (IC 95%: −243.94, 1056.81), while the more extreme scenario of a 15% reduction may lead to 2086.62 fewer cases (IC 95%: 1426.66, 2746.57). On the other hand, if demand contraction in the processed meat sector resulted in a 0.1% loss in employment, one could expect 27.23 all-cause mortalities attributable to job loss (IC 95%: 16.55, 37.80). This simulation study demonstrates that caution should be taken when implementing public awareness campaigns, particularly when the prevention message is not straightforward.

## 1. Introduction

In recent decades, much attention has been given to providing effective communication strategies regarding nutritional risk, with the aim to avoid unnecessary alarm to consumers [1,2]. Effective risk communication has become rather challenging, with the most recent example being food-related concerns about the risk of colorectal cancer (CRC) associated with the consumption of red and processed meat [3,4]. Red meat consumption, both raw and processed, has been associated with several diseases, from diabetes mellitus to coronary heart disease [5] and cancer [6,7,8]. Nevertheless, this has become a sensitive topic, especially after the World Health Organization (WHO) classified processed meat as Group 1 (carcinogenic to humans) and red meat as Group 2A (probably carcinogenic to humans) [3,4]. Mass media immediately disseminated the WHO announcement, which was widely cited in the most influential press agencies and commentaries.

There is no doubt that the WHO recommendation about safe levels of processed meat intake is sound and based on empirical evidence. WHO classified “processed meat as carcinogenic to humans (Group 1), based on sufficient evidence in humans that the consumption of processed meat causes colorectal cancer”. This claim was based on more than 800 studies involving more than a dozen cancer types, using data from different countries and accounting for different diets and lifestyles [3,4]. However, the mechanism through which their message reached the consumers was distorted, with potential unforeseen direct and potentially harmful indirect consequences. Communication of the carcinogenicity of processed meat consumption was made official in the WHO–IARC (International Agency for Research on Cancer) press release, disseminated at the public opinion level through an intense campaign aimed at raising awareness on the risks associated with red and processed meat consumption. The core of the WHO communication strategy has been structured around the “level of evidence” of the association between processed meat consumption and CRC, which is the reason for the Group 1 classification [3,4]. Group 1 includes agents for which there is sufficient evidence of carcinogenicity in humans. However, the groupings are constructs that refer to the level of evidence, not to the magnitude of risk of neoplasm related to the agent under study. Groupings are based on strict criteria of evidence of causal association between the agent and malignant neoplasms. However, the magnitude of such causal association is not included in the criteria for including an agent in Group 1. For example, two different agents, both included in Group 1, are agents for which there is sufficient evidence of association with neoplasms, but the magnitude of association could be completely different for each. 

The interpretation of such concepts is not straightforward, and we cannot rule out that the concept of “level of evidence” has been mistaken for that of “magnitude of association”. Table 1 shows some extracts from selected influential international newspaper headlines and news reports showing different degrees of bias in the interpretation of the WHO announcement.

Regarding the actual figures that are the basis of the discussion, IARC estimated that every 50-g portion of processed meat eaten daily increases the risk of CRC by approximately 18% (1.18, 95% CI 1.10–1.28) [3,4]. However, the estimated relative risk of current smoking (which belong to the same group as processed meat) for squamous cell cancer in men is 45.6 (95% CI 34.3–60.6) [9]. The levels of risk are completely different in terms of both individual and general public health impact [10]. 

The relationship between media influence and consumer reaction is not a new topic in sociology. In a seminal paper [11], their interaction was characterized in five common steps: (i) spreading sensational news about the potential risk factor; (ii) public concern increases along with the media coverage; (iii) public response begins, usually with a shift in consumption patterns; (iv) the alarm gradually decreases as attention switches away from the issue, leading to a new equilibrium. Additionally, it is widely recognized that even risk events with minor physical consequences often elicit strong public concern and produce extraordinarily severe social impacts [12]. These two theories, when applied in the case of the WHO announcement, suggest that even though the risk of cancer is real, the lack of careful risk communication strategy might have unintended negative impacts [2]. We have already learned from previous food scares that consumers do react and modify their patterns and preferences, often in a sharp manner [13]. For example, in 1996 there was an announcement by the WHO regarding Creutzfeldt–Jakob disease in humans and bovine spongiform encephalopathy (BSE) in cattle. This resulted in enormous media attention and had an immediate negative impact on the UK beef industry [14]. This was reflected in actual consumer behavior, where beef consumption dropped by 17% in 1996, a 10% greater decrease than expected [14,15].

Food scares shape consumption preferences, even when there is no sufficient evidence about the real causal mechanism between the contaminant and health risk. Genetic modification (GM) of food has been given a great deal of media attention (in both the United Kingdom and in Europe more generally), particularly in early 1999 [16]. The WHO states that the GM foods currently available on the international market have passed safety assessments and are unlikely to present risks for human health. In addition, no effects on human health have been shown as a result of the consumption of such foods by the general population in the countries where they have been approved. However, in 1998 more than 1000 UK schools took genetically modified foods off their menus, and genetically modified foods were banned from restaurants and bars in the House of Commons. Furthermore, most of the major UK supermarkets have eliminated genetically modified ingredients from their own brand products in response to consumer concern [16]. Similarly, despite the benefits of reducing red and processed meat consumption, the adoption of an aggressive communication strategy could lead to a wide range of scenarios, which should be carefully taken into account. 

In this paper, we explore the potential counter-effects of such communication campaigns, raising the idea of a potential balloon effect, reflected in sectorial economic shock and contraction of sales as a foreground for future public health implications. Indeed, the literature suggests that, depending on the demand elasticity, processed meat stigmatization might induce a significant shock to the industry and lead to the displacement of workers, temporary unemployment, and instability, which all potentially lead to adverse health outcomes.

Our analysis combines real data and parameters found in recent literature to simulate a realistic scenario highlighting the potential benefits and costs of a WHO-announcement-solicited food scare. We simulated the consequences of risk communication to analyze the impacts of a reduced consumption of processed meat on the yearly number of cancers, juxtaposed with the adverse health effects deriving from job loss/displacement due to the sectorial economic shock caused by a shift in demand pattern.

## 2. Materials and Methods 

In view of analyzing the interface between the impacts of reduced processed meat consumption on CRC cases and on the meat industry, two simulation scenarios were set up—one on the effect of reduced processed meat consumption on CRC cases, and the other on job loss impacts on the health outcomes of meat industry workers (assuming that the processed meat consumption reduction resulted in a reduced demand for processed meat). 

Monte Carlo microsimulations were performed using Model Risk [17]. Ten thousand simulations were run on a series of stochastic models that were built around data derived from the literature. The incidence and mortality rates were modeled according to binomial distributions. The final results are presented using 95% credibility intervals. 

The sections below present the framework and the data used for the two simulation scenarios.

### 2.1. Meat Consumption and Colorectal Cancer

Data about processed meat consumption and CRC incidence statistics were derived from The European Prospective Investigation into Cancer and Nutrition (EPIC) study [18]. The EPIC study was one of the largest European cohort studies, with almost half a million people, designed to examine the relationship between diet and cancer. It involves 10 EU countries: the UK, the Netherlands, Sweden, Spain, Norway, Italy, Greece, Germany, France, and Denmark. We decided to consider EPIC’s structure and results because of its detailed information about processed meat consumption levels across different regions in the EU, the presence of CRC incidence statistics, and its large sample size. 

The EPIC population was divided according to the level of processed meat consumption as follows: the lowest (<10 g/day), medium-low (10–20 g/day), medium (20–40 g/day), medium-high (40–80 g/day), and highest (>80 g/day). From 1992 to approximately 2002, the total number of person-years was 2,279,075. The number of CRC cases for each consumption group is given in Table 2. We chose to focus on only processed meat and CRC, since other types of cancer are not supported by the same level of robust empirical evidence. The distribution of processed meat consumption was obtained using 24 hour recall survey data (Table 2).

### 2.2. Job Loss and Health Consequences

We simulated the potential adverse health effects due to job loss/displacement in meat industry workers. The European meat industry represents a large economic sector, with 44,000 enterprises worth 30 billion dollars and employing 1 million people, of which 54.2% are in the processed meat industry [19]. Since no empirical evidence is available to predict what would be the revenue loss in each case of reduced processed meat consumption, data about similar industries that have undergone consumption shocks or aggressive demand contraction policies [20,21,22] were used for simulation purposes. Therefore, we have assumed 4 hypothetical scenarios of job losses, from a minimum of 0.1% (as in the Chinese tobacco taxation case [20]) to 1% (as in the Canadian BSE case [23]). We used meat production statistics [24] to simulate the number of potential displaced workers from the EU processed meat industry. For each employment scenario, the corresponding long-term health effects were estimated using the Eliason and Storrie study [25] on the effect of closure of plants on the mortality of displaced workers. We chose this study since it does not suffer from reverse causality, which is often encountered when dealing with the health effects of unemployment [26,27,28]. The reverse causality problem refers to the fact that it is difficult to establish whether poor health is a consequence of the job loss or whether there was selection bias, where illness-prone workers were more likely to be laid off. In the selected study, the dismissal of the workers was determined by an external factor (plant closure), so the reverse causality problem was avoided. 

Using the all-cause and cause-specific mortality probabilities from this study, we simulated the expected number of deaths in the four years after the job loss event.

## 3. Results

### 3.1. Meat Consumption and Colorectal Cancer

The Monte Carlo estimate of baseline new yearly cases of CRC in the 10 EPIC countries was 289035.14 (95% CI 192679.99–359491.33). This was the starting point to simulate the benefits (in terms of a reduced number of CRC cases) of restricted processed meat intake (Table 3). Based on demographic statistics from 2010, our model suggests that a 1% reduction in the number of high-level consumers of processed meat (>40 g/day) would not lead to a statistically significant decrease in CRC cases (95% CI: −243.94–1056.81). Significant effects were associated with a more diffuse reduction in consumption patterns, such as 5%, 10%, and 15% reductions, which would respectively lead to reductions of 1010.28, 1729.23, and 2086.62 in the incidence of CRC.

### 3.2. Job Loss and Health Consequences

The unintended consequences of restricting processed meat consumption were analyzed by simulating the effect of job loss on adverse health effects of meat industry workers. We assumed that the changing consumption habits might result in decreased sales, and the consequent decline of several industries, as has been demonstrated in recent incidents [20,21,23,29,30]. This is of particular concern in markets such as the EU, with a large labor market and institutional rigidity [31]. The simulation results are reported in Table 4. Using the 2010 processed meat industry fact sheet, in the scenario in which an awareness campaign leads to substantial demand contraction, resulting in a 0.1% loss in employment in the processed meat sector (best case scenario), one could expect a 27.23 (IC 95%: 16.55, 37.80) all-cause mortality attributable to job losses in the meat sector. In the worst-case scenario of a sharp market reaction, reflected in a 1% employment loss, the number of deaths increases to 272.30 (IC 95%: 165.53, 378.02). Specifically, tumor-related mortality ranges across the scenarios from approximately 10 (−1.12, 20.13) to one hundred (−11.24, 201.25), with similar results for deaths due to circulatory diseases (from 8.76 (IC 95%: −1.92, 19.33) to 87.60 (IC 95%: −19.18, 193.32)). Death due to suicide ranges from 2.14 for the best case scenario of 0.1% employment loss (IC 95% −9.19, 12.06) to 21.35 in the worst-case scenario of 1% employment loss (IC 95% −91.86, 120.63). Alcohol-related mortality ranges from 1.49 (IC 95% −8.54, 12.71) to 14.91 (IC 95% −85.42, 127.07).

## 4. Discussion

The aim of the present study was to analyze, through a simulation approach, the benefits of reduced processed meat consumption on CRC cases and the potential impact of such a reduction on the processed meat market, in terms of adverse health effects deriving from job loss/displacement. The hypothesis underlying the study was that sensational communication about the risk associated with processed meat consumption would have severe unintended effects on the health outcomes of the workers employed in this sector due to a potential market shock, as demonstrated by previous studies in the field [9,22,28,32,33,34,35,36]. An example of this situation is the case of BSE, which strongly impacted the world meat economy in the 1990s. In countries with meat-intensive production, such as Northern Ireland, 5135 full-time job equivalents were threatened by a reduction in demand for beef, accounting for approximately 0.6% of total regional employment [9,22,32]. The literature also shows that these types of scenarios may have important public health issues. In the short term, the most significant burden lies in the costs associated with mental health, such as the purchase of psychotropic drugs and hospitalizations due to mental health problems [28] and hazardous behavior (alcohol and illegal substance abuse) [33]. Hazardous behavior due to job loss, excessive stress, smoking, and alcohol consumption are linked to circulatory disease and hospitalization due to traffic accidents [37]. Smoking is strongly associated with lung cancer and several other cancers [9], and both psychological distress and alcohol abuse are associated with suicide [25,33]. In the long term, the effects of job loss and unemployment can result in an increased mortality, especially for older workers. Indeed, mortality rates in the year after displacement are 50%–100% higher than would otherwise have been expected, and this effect persists even after 20 years [34]. Another recent meta-analysis has assessed the association between unemployment and all-cause mortality among working-age persons, and they found that unemployment is associated with a 63% higher risk of mortality [35]. Hence, it would not be unlikely that, within the framework of a persistent economic crisis [36], a contraction in the demand for red and processed meat might lead to a reduced market share of the processed meat industry, with subsequent long-term effects on public health. 

The hypothesis about the potential association between CRC and processed meat consumption is a matter for criticism for two main reasons. The first one is represented by the risk communication modalities themselves, which could have been misleading. Risk communication has been extensively studied due to its critical impact on the general population [2]. It represents a complex issue influenced by several factors, including characteristics of the public, of the message itself, and trust of the institution communicating the risk [1,38]. It was not the purpose of the present work to provide an excuse of the theories of risk communication and its related issues. However, it is essential to point out a few critical elements that seem to be involved in the risk communication of the carcinogenicity of red and processed meat, forming the basis of the present study. It has been shown that several mechanisms may be involved in producing unintended effects of risk communication campaigns. An example of such mechanism is the phenomenon of “social amplification risk” [39]. This is a complex phenomenon involving different social agents. Briefly, when a new research finding is published, mass media communicate it to the public. However, the risk representation of the researchers might be not the same as that of the mass media, which may exaggerate the risk when communicating it to the public to make the scientific finding appear more newsworthy [40,41]. This process may lead to secondary, unintended effects [39,42]. The second reason for criticism is that the WHO-IARC communication was based on research that would normally be subject to criticism and revision, as has been demonstrated by the recent work of Kruger and Zhou [43]. Such work highlighted the fact that the IARC claimed that the association between CRC and red meat is supported by strong mechanistic evidence, citing heme as one red meat component responsible for the carcinogenic process. However, Kruger and Zhou concluded that studies in the field do not provide sufficient evidence about the role of heme on the risk of CRC.

The results of the present study highlight that there might be a substantial interaction between the potential benefits of an awareness campaign in terms of reducing CRC incidence and the potential long-term costs encountered by an important industrial sector in terms of increasing mortality due to job losses and other unintended economic consequences. Considering this framework and the results provided by the simulation study, it is essential to carefully plan risk communications, taking into account their potential psychological and social implications, since they may result in secondary consequences that may have a negative impact on the market [1]. 

Clearly, the aim of WHO was to provide material for consumers to help them make informed decisions regarding their food choices. However, separating the concepts of level of evidence (a complex epistemological construct) from that of the magnitude of the risk (which is a complex probabilistic concept referring to the size of the effect of the agent on the outcome of interest) is a difficult cognitive exercise [44]. Understanding these concepts and separating them (since they are independent from each other) requires specific training. Thus, the most likely understood message by the average consumer was, potentially, that eating processed meat was “as dangerous as smoking”. resulting in a misleading risk representation. The risk misrepresentation (and exaggeration) could reasonably lead to more extreme decisions, such as eliminating eating processed meat altogether, instead of the intended ones, such as a conscious reshaping of eating habits and individual lifestyle choices. Even though this represents a simple setting simulation study, it should make us reflect upon the need to carefully identify an optimal approach to communicate nutritional risks. Indeed, if the perception of the producers and consumers is that the level of risk of processed meat is comparable to that of smoking or asbestos exposure, it would not be surprising that some consumption pattern reshaping might happen. In Europe, which still suffers from a major economic crisis and which is slowly but consistently moving towards a more flexible labor market, some markets might face a marked decline. The literature shows that market shocks and job losses do not come without adverse consequences to individuals’ health and their propensity for risky behaviors. 

To overcome these issues, the development of risk communication guidelines for media professionals has been suggested [45] to promote clear risk communication, so that secondary consequences can be avoided. Because many newspapers reported that the classification of processed meat was in the same group as tobacco and asbestos, we argue that the meaning of this message might not have been properly assimilated by the general public. It is clear that even if these two factors are found to be carcinogenic on the basis of analogously strong and consistent empirical evidence, the levels of risk have completely different repercussions on overall public health. 

What type of consequences could we imagine in this context? Although we do not have any empirical constructs that could provide a robust answer to this question, we simulated possible outcomes using findings from recent literature. Our scenarios are hypothetical but reasonable and show an important balance between the benefits of a lower CRC incidence and the costs in terms of adverse health effects related to a potential market shock for such a large industry. What makes a society better off? A gain of 406 yearly saved CRC cases is definitely a good. However, what if that gain generates some medium-term externalities that may result in 27 deaths attributable to job loss?

## 5. Conclusions

The interconnectedness between different segments of society in such a long-term scenario remains speculative. In an ideal world, one should prevent adverse health impacts, as well as any adverse socioeconomic consequences of interventions, and a correct knowledge and interpretation of the “magnitude of risk” by all stakeholders is of primary importance. 

The scenario here simulated suggests caution when performing large-scale, important health risk communications, since inappropriate modalities might generate undesirable side effects and related consequences. This is particularly true when the magnitude of risk is modest, as is the case for most nutritional risk factors, and when it is not immediately and completely clear whether the overall benefits will be higher than the potential costs.

We emphasize that risk communication represents an important part of risk management. Accordingly, all the potential externalities should be examined and considered before launching large-scale and sensationalistic awareness campaigns, given the potential for the unintended consequences of such campaigns.

## Figures and Tables

**Table 1 nutrients-11-00400-t001:** Clippings from international media press around 26 October 2015 regarding the WHO announcement about the carcinogenicity of red and processed meat.

Media	Date	News
The Telegraph	26 October 2015	*“Processed meat ranks alongside smoking as major cause of cancer, World Health Organisation say.”* *“WHO publishes report listing processed meat as ‘carcinogenic to humans’—the highest ranking, shared with alcohol, asbestos, arsenic and cigarettes”*
BBC news	26 October 2015	*“Processed meats—such as bacon, sausages and ham—do cause cancer, according to the World Health Organization (WHO).”* *“It has now placed processed meat in the same category as plutonium, but also alcohol as they definitely do cause cancer.* *However, this does not mean they are equally dangerous. A bacon sandwich is not as bad as smoking.”*
NBC news	26 October 2015	*“Most reports on the links between meat and cancer have been softened with some element of doubt, but the IARC uses clear and direct language in saying processed meat causes cancer. There are no phrases such as "may cause" in the report.”*
The Times of India	26 October 2015	*“The France-based International Agency for Research on Cancer (IARC), part of the WHO, put processed meat like hot dogs and ham in its group 1 list, which already includes tobacco, asbestos and diesel fumes, for which there is "sufficient evidence" of cancer links.”*
Al Jazeera America	27 October 2015	*“The World Health Organization’s (WHO) announcement, Monday, that eating processed meats significantly raises a person’s cancer risk is bad news for low-income households, which consume a lot of such food, research shows.”*
Financial Times	25 October 2015	*“The global meat industry has reacted with fury to the prospect of its products being declared carcinogenic by the cancer research arm of the World Health Organisation (…) In an attempt to undermine the IARC findings in advance, The North American Meat Institute (NAMI) accused the body of “dramatic and alarmist overreach.”*
Reuters Italy	27 October 2015	*“Italian food and farming groups responded indignantly to the World Health Organization (WHO) report that put cured meats, such as ham, sausage and salami, together with asbestos and tobacco on a list of carcinogens.* *“No to meat terrorism, the Italian stuff is the healthiest,” agricultural association Coldiretti said in a statement, crediting the country’s diet for one of the highest life expectancies in the world—80 years for men and 85 for women.”*
Reuters Germany	27 October 2015	*“The report, which classified processed meat as “carcinogenic to humans” on its group one list along with tobacco and asbestos, drew attention in Germany, the world’s highest consumer of such products.* *“No one should be afraid if they eat a bratwurst (sausage) every now and then,” Christian Schmidt, minister for food and agriculture, said in a statement emailed to Reuters.* *“People are being wrongly unsettled when eating meat is put on the same level as asbestos or tobacco.””*
The Irish Times	30 October 2015	*“WHO: Processed meat cancer report message ‘misinterpreted’* *Mr* *Härtl (IARC) said it was a “shortcoming” of the classification system that tobacco, processed meats and arsenic were in the same group.* *“We do not want to compare tobacco and meat because we know that no level of tobacco is safe,” he said.”*
Global News		*“The Canadian Cancer Society said when it comes to processed meat, Canadians should, “limit your intake and consider it only for things like special occasions” said Sian Bevan, director of research for the Canadian Cancer Society. “Holidays, birthdays… and at a special sporting event. It’s about not making it a daily choice or a regular choice.”* *This has been the cancer society’s recommendation since 2009—avoid processed meat.”*

Group 1 has been defined as a cluster “which already includes tobacco, asbestos and diesel fumes” (The Times of India), alluding to the similar impact on human health. Furthermore, potential individual-level benefits were freely hypothesized upon several scenarios of intake reduction, which range from “limit your intake and consider it only for things like special occasions” to “avoid processed meat” (Global News). The Telegraph states that “Processed meat ranks alongside smoking as major cause of cancer”; BBC news explains that “It has now placed processed meat in the same category as plutonium, but also alcohol as they definitely do cause cancer”. The BBC also acknowledges that the level of danger is not the same; however, the first message absorbed is about clustering processed meat with widely known high-risk hazards, such as smoking, plutonium, or alcohol. On the other hand, other media outlets, such as Al Jazeera America and the Financial Times, warn about the alarmist dimension of the message, and Reuters Germany and the Irish Times clearly express the misinterpretation of the main WHO message.

**Table 2 nutrients-11-00400-t002:** European Prospective Investigation into Cancer and Nutrition (EPIC) study data of colorectal cancer and associated processed meat consumption (Norat et al., 2005 [7]).

Consumption (g/day)	N cases of Colorectal Cancer
<10	232
10 to 20	256
20 to 40	402
40 to 80	318
≥80	121

For simplicity, we assumed that only high-level consumers of processed meat (>40 g/day) would drastically change their behavior, while the low-level consumers would maintain their consumption. These assumptions are reasonable because the WHO states that a consumption >50 g/day represents a hazard. We hypothesized 4 different scenarios of processed meat consumption decrease: low (−1%), moderate (−5%), and relatively high (10% and 15%). The conservative choice of the percentages reflects the assumption that no radical choices would be made in terms of diet preferences. For each meat reduction scenario, we simulated the person-year reduction in CRC incidence using the cancer probabilities derived from the EPIC study.

**Table 3 nutrients-11-00400-t003:** Monte-Carlo simulation results: impact of reduced processed meat consumption on the reduction of colorectal cancer (cases).

	Processed Meat Reduction Scenarios			
	−1%	−5%	−10%	−15%
Colorectal (CR) cancer incidence reduction	406.43	1010.28	td align="center" valign="middle" style="border-bottom:solid thin">1729.23	td align="center" valign="middle" style="border-bottom:solid thin">2086.62
C.I. 95%	(−243.94, 1056,81)	(358.04, 1662.52)	(1073.43, 2385.03)	(1426.66, 2746.57)

**Table 4 nutrients-11-00400-t004:** Monte-Carlo simulation results: impact of job loss on different causes of mortality, in terms of expected number of deaths in 4 years following the job loss.

	Employment Loss Scenario			
	−0.1%	−0.3%	−0.6%	−1%
All-cause				
Expected deaths	27.23	81.69	163.38	272.30
95% C.I.	(16.55, 37.80)	(49.66, 113.41)	(99.32, 226.81)	(165.53, 378.02)
Malign neoplasm				
Expected deaths	9.55	28.66	57.32	95.53
95% C.I.	(−1.12, 20.13)	(−3.37, 60.38)	(−6.74, 120.75)	(−11.24, 201.25)
Circulatory diseases				
Expected deaths	8.76	26.28	52.56	87.60
95% C.I.	(−1.92, 19.33)	(−5.75, 58.00)	(−11.51, 115.99)	(−19.18, 193.32)
Suicide				
Expected deaths	2.14	6.41	12.81	21.35
95% C.I.	(−9.19, 12.06)	(−27.56, 36.19)	(−55.12, 72.38)	(−91.86, 120.63)
Alcohol-related mortality				
Expected deaths	1.49	4.47	8.95	14.91
95% C.I.	(−8.54, 12.71)	(−25.63, 38.12)	(−51.25, 76.24)	(−85.42, 127.07)

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
