# Peer review of "Communicating Risk Regarding Food Consumption: The Case of Processed Meat"

_nutrients, 2019, doi:10.3390/nu11020400_

Round 1

Reviewer 1 Report

The topic of the article seems very interesting. However, the authors do not justify their hypotheses, in fact, in the conclusions they comment that they can not give an answer to the questions that society ask.

In addition, the results and discussion parts are very brief and focus a lot on comparing the risk of meat consumption with the loss of work. In this sense, Table 5 focuses on impact of job loss on different causes mortality (all-causes, malign neoplasm, circulatory disease, suicide, alcohol). It seems that nothing has to do with what the work proposes.

Moreover, the discussion of the results showed in Tables 3 and 4, related to the effect of processed meat consumption on reduction in colorectal cancer, is hardly justify. Trivial comments are made, mentioning even topics such as BSE, and public health issues as mental health problems.

Author Response

Clarifications have been added to the methods, results, and discussion sections according to reviewer’s requests. In addition to that, the manuscript underwent a certified revision of the English language.

Comments and Suggestions for Authors

1)      The topic of the article seems very interesting. However, the authors do not justify their hypotheses, in fact, in the conclusions they comment that they can not give an answer to the questions that society ask.

Clarifications about the hypotheses underlying the study have been provided in the Discussion and relevant references have been added (“Gallus, S.; Bosetti, C. Meat consumption is not tobacco smoking. International Journal of Cancer 2016, 138, 2539–2540.” and “Kruger, C.; Zhou, Y. Red meat and colon cancer: A review of mechanistic evidence for heme in the context of risk assessment methodology. Food and chemical toxicology 2018.”).

2)      In addition, the results and discussion parts are very brief and focus a lot on comparing the risk of meat consumption with the loss of work. In this sense, Table 5 focuses on impact of job loss on different causes mortality (all-causes, malign neoplasm, circulatory disease, suicide, alcohol). It seems that nothing has to do with what the work proposes.

The Discussion has been improved and clarifications about the role of Table 5 (now Table 4) have been added to the Methods and the Results.

3)      Moreover, the discussion of the results showed in Tables 3 and 4, related to the effect of processed meat consumption on reduction in colorectal cancer, is hardly justify. Trivial comments are made, mentioning even topics such as BSE, and public health issues as mental health problems.

The Discussion has been improved as requested. In addition to that, we have rearranged the Tables to allow for a better understanding of the simulation results.

Reviewer 2 Report

General comments

The article is presents valuable information.

1.      ‘IARC’ needs to be expanded in the beginning of the manuscript.

Specific

Line 14: ‘’allow for a setting up of a realistic scenario….” Wrong sentence structure

Line 16-17: “Impact of communication on risks of processed meat intake were….”

Change to ‘was’ (‘Impact’ is a singular)

Line 34: “World Health Organization (WHO) announcement that there processed meat has been…..”

Remove ‘there’

Line 39: “WHO recommendation about safe levels of processed meat intake are sound and based on strong empirical evidence.”

Change to “is” (‘Recommendation’ is singular)

Line 44-45: “was a sort of a distortion mirror..’

Change to ‘Was sort of a distortion mirror..’

Line 63-67: “Group 1 is based on strict criteria of evidence of causal association between the agent and malignant 63 neoplasms. However, the magnitude of such causal association is not included in the criteria for 64 including an agent in the Group 1. Two agents, both included in the Group 1, are both agents for 65 which there is sufficient evidence of association with neoplasms, but the magnitude of association 66 could be completely different.”

Repetition of Lines 54-59. Remove these lines.

Line 68-69: “The interpretation of such concepts is not straightforward, and we cannot rule that the concept 68 of “level of evidence” has been mistaken with that of “magnitude of association” (see Table 1).”

Repetition of lines 60-61. Remove these lines.

Line 71: “alluding on..”

Right usage would be “alluding to”

Line 80: “such as smoking, plutonium or alcohol”

Missing comma after plutonium. Punctuation mistake.

Lines 80-83: “On the other hand, some other media, such as Al Jazeera America or Financial Times, warn about the alarmist dimension of the message diffused, while media such as The Reuters Germany or the Irish Times clearly express the misinterpretation of the main WHO message”

Restructure the sentence.

Lines 91-92: “public concern increases as the along with the media coverage”

Change the highlighted part

Author Response

The article is presents valuable information.

1.          ‘IARC’ needs to be expanded in the beginning of the manuscript.

Done

Specific

1)      Line 14: ‘’allow for a setting up of a realistic scenario….” Wrong sentence structure

Done

2)      Line 16-17: “Impact of communication on risks of processed meat intake were….”

Change to ‘was’ (‘Impact’ is a singular)

Done

3)      Line 34: “World Health Organization (WHO) announcement that there processed meat has been…..”

Remove ‘there’

Done

4)      Line 39: “WHO recommendation about safe levels of processed meat intake are sound and based on strong empirical evidence.”

Change to “is” (‘Recommendation’ is singular)

Done

5)      Line 44-45: “was a sort of a distortion mirror..’

Change to ‘Was sort of a distortion mirror..’

Done

6)      Line 63-67: “Group 1 is based on strict criteria of evidence of causal association between the agent and malignant 63 neoplasms. However, the magnitude of such causal association is not included in the criteria for 64 including an agent in the Group 1. Two agents, both included in the Group 1, are both agents for 65 which there is sufficient evidence of association with neoplasms, but the magnitude of association 66 could be completely different.”

Repetition of Lines 54-59. Remove these lines.

Done

7)      Line 68-69: “The interpretation of such concepts is not straightforward, and we cannot rule that the concept 68 of “level of evidence” has been mistaken with that of “magnitude of association” (see Table 1).”

Repetition of lines 60-61. Remove these lines.

Done

8)      Line 71: “alluding on..”

Right usage would be “alluding to”

Done

9)      Line 80: “such as smoking, plutonium or alcohol”

Missing comma after plutonium. Punctuation mistake.

Done

10)   Lines 80-83: “On the other hand, some other media, such as Al Jazeera America or Financial Times, warn about the alarmist dimension of the message diffused, while media such as The Reuters Germany or the Irish Times clearly express the misinterpretation of the main WHO message”

Restructure the sentence.

Done

11)   Lines 91-92: “public concern increases as the along with the media coverage”

Change the highlighted part

Done

Round 2

Reviewer 1 Report

The manuscript has been improved. I appreciate the effort of authors to explain better the results of the manuscript.